# Sleep Quality among Patients with Type 2 Diabetes: A Cross-Sectional Study in the East Coast Region of Peninsular Malaysia

**DOI:** 10.3390/ijerph19095211

**Published:** 2022-04-25

**Authors:** Nor Fareshah Mohd Nasir, Nani Draman, Maryam Mohd Zulkifli, Rosediani Muhamad, Samsul Draman

**Affiliations:** 1Department of Family Medicine, School of Medical Sciences, Universiti Sains Malaysia, Kubang Kerian 16150, Malaysia; fareshah@student.usm.my (N.F.M.N.); maryammz@usm.my (M.M.Z.); rosesyam@usm.my (R.M.); 2Department of Family Medicine, Kulliyyah of Medicine, International Islamic University Malaysia, Jalan Sultan Ahmad Shah, Kuantan 25200, Malaysia; nurin@iium.edu.my

**Keywords:** type 2 diabetes, sleep quality, diabetes distress, quality of life

## Abstract

Poor sleep is related to type 2 diabetes and adversely influences a person’s quality of life. This study aimed to evaluate sleep quality in patients with type 2 diabetes (T2DM), its associated factors, and its relationship with quality of life. A cross-sectional study was conducted at a primary care clinic in a tertiary hospital on the east coast of Malaysia. This study included 350 participants (175 men and 175 women). Data were collected using the Malay version of the Pittsburgh Sleep Quality Index (PSQI-M) with a cut-off point of >5 as poor sleep, the Malay version of Diabetes Distress Scale (MDDS-17) and the revised Malay version of T2DM-related quality of life (Rv-DQOL). Statistical analysis was conducted using the SPSS software version 26.0. The respondents’ median (interquartile range (IQR)) age was 62.0 (11.0) years, and poor sleep was reported in 32% (95% confidence interval (CI) = 27.1, 36.9) of the participants. Multivariate logistic regression analysis revealed that poor sleep quality was significantly associated with nocturia (odds ratio (OR) = 2.04; 95% CI = 1.24, 3.35), restless legs syndrome (OR = 2.17; 95% CI = 1.32–3.56) and emotional burden (OR = 2.37; 95% CI = 1.41–3.98). However, no statistically significant association was observed between sleep quality and quality of life among our participants.

## 1. Introduction

The prevalence of type II diabetes (T2DM) is increasing worldwide. The latest edition of the International Diabetes Federation (IDF) Diabetes Atlas (2021) reported that T2DM affects 537 million people globally. It is expected that by 2030, this number will increase to 643 million. Currently, 90 million individuals are living with T2DM in Southeast Asia, and the number is anticipated to increase by 63% to 130 million by 2045. IDF declared T2DM as one of the fastest-growing worldwide health emergencies of the twenty-first century [1]. The complications are debilitating and have contributed to rising morbidity and mortality rates. T2DM is responsible for 1.6 million deaths each year worldwide, and other negative health outcomes include stroke, heart disease, hearing loss, blindness, hypertension, renal disease and nerve damage [2].

According to growing evidence, sleep deprivation has been linked to an increased risk of T2DM. Sleep plays a vital role in modulating endocrine, metabolic function and sympathovagal balance, and its deprivation has negative impacts on these systems [3]. Studies conducted in Albania have shown poor sleep quality as a result of disruption in circadian rhythm, e.g., night shift work among nurses causes a significant increase in HbA1c and body mass index (BMI) and influences certain gene expression in peripheral blood mononuclear cells that act as “peripheral clock genes”. These changes typically manifest in people at risk of developing chronic metabolic diseases, especially T2DM [4,5].

Conversely, people with T2DM have a more significant risk of having sleep disorders than the general population [6]. Sleep and wakefulness problems are common in T2DM due to physiological imbalances and co-morbid sleep disorders, which result in poor sleep quality [7]. Obstructive sleep apnoea (OSA) is a common concomitant sleeping disorder in T2DM, especially among obese patients. The severity of OSA is linked to poor glucose control [8].

T2DM complications, such as painful diabetic polyneuropathy, restless legs syndrome (RLS), nocturia and sleep-related breathing disturbance, can impair sleep initiation and maintenance, resulting in decreased sleep quality, daytime fatigue and somnolence [9]. A recent meta-analysis revealed that T2DM patients with poor sleep quality have greater HbA1c levels. There is a dose–response U-shaped relationship between sleep duration and HbA1c; short and long sleep durations had detrimental effects on HbA1c compared with normal sleep duration [8]. This indicates how critical adequate sleep is for T2DM patients.

Sleep quality is a measure of a person’s sensation of being healthy, active, and ready for a new day, whereas adequate sleep is defined as a multidimensional construct rather than a single dimension, such as duration [10]. Sleep quality is objectively evaluated using instruments such as the actigraph, polysomnograph and sleep electroencephalogram to determine sleep patterns [11]. Subjectively, sleep quality can be measured using various validated questionnaires addressing multiple aspects of sleep construct, such as latency and efficiency, and among the widely used tool to measure sleep quality is the Pittsburgh Sleep Quality Index (PSQI) [12].

In the medical domain, quality of life is expressed as ‘subjective health’ or ‘functional status and well-being’. It represents the effect of an illness as perceived by the subjects and is often used as an outcome measurement [13]. People with T2DM generally have a worse quality of life compared with healthy people. It is found that complications of T2DM are a significant disease-specific determinant of quality of life [14]. Sleep-related disturbances have been associated with a more inferior quality of life among T2DM patients [15,16].

Primary care providers are frequently the first point of contact for T2DM patients. Education on the importance of sleep should begin as soon as a patient is diagnosed, focusing on disease understanding and treatment compliance. Thus, efforts should be made to raise awareness of the relationship between sleep quality and illness. In Malaysia, there is a paucity of reporting on sleep quality associates among patients with T2DM. Thus, we conducted a cross-sectional study to look at the prevalence of poor sleep quality among patients with T2DM, its associated factors and its relationship with the quality of life.

## 2. Materials and Methods

### 2.1. Participants

This is a cross-sectional study conducted at Klinik Rawatan Keluarga, a primary care clinic at the Hospital Universiti Sains Malaysia, from November 2020 to January 2021. The participants were T2DM patients attending regular follow-up. Those aged more than 18 years old were included in this study. Those who suffered from acute complications of T2DM, such as hypoglycaemia, diabetic ketoacidosis and hyperglycaemic hyperosmolar coma, were excluded from this study. We also excluded those having other severe comorbidities, such as heart failure (uncontrolled with medications), kidney failure (stage 5 and end-stage renal failure) and respiratory failure (such as co-morbid chronic obstructive pulmonary disease (COPD)), which would significantly affect the quality of life.

### 2.2. Measures

The sample size was calculated based on comparing two proportions for categorical variables using PS software version 3.1.2 with the power of study set at 80% and type I error set at 5%. Based on a study by Bani-Issa et al. (2018), the calculated sample size was 318, and after considering 10% non-response, the final sample size was 350 [17].

The clinical research tool used in this study consisted of a case report form. The case report form consisted of six sections. Section 1 included socio-demographic information. Section 2 provided details on DM among participants inclusive of the duration of diagnosis and treatment. The second part of Section 2 consists of self-reported symptoms of three T2DM complications, namely, nocturia, RLS and diabetic neuropathy. Section 3 provided information on the patient’s height (m), weight (kg) and HbA1c (%).

Section 4 was the Malay version of the Pittsburgh Sleep Quality Index (PSQI-M). This self-administered questionnaire was originally developed by Buysse et al. in 1989, which was primarily intended to discriminate between poor and good sleepers over the past 1 month time [18]. This was translated by Farah et al. (2019) with acceptable internal consistency values of 0.27 (test) and 0.68 (retest) [12]. The PSQI consists of 19 items; it is a self-report questionnaire that measures subjective sleep quality. These items are aggregated into seven components: (1) sleep quality (1 item), (2) sleep latency (2 items), (3) sleep duration (1 item), (4) sleep efficiency (3 items), (5) sleep disturbance (9 items), (6) sleep medication (1 item) and (7) daily dysfunction (2 items). Each component has a score ranging from 0 to 3; the scores are summed up to yield a PSQI global score ranging from 0 to 21. Respondents with a score of greater than 5 are ‘poor sleepers’, whereas those with a score of 5 or less are ‘good sleepers’ [18].

Section 5 consisted of the Malay version of T2DM distress scale (MDDS-17) translated by Chew et al. (2015) adapted from the original 17-item Diabetes Distress Scale by Polonsky et al. (2005), which assessed related problems and hassles concerning T2DM patients for the past 1 month using a Likert scale from 1 (not a problem) to 6 (a very serious problem). Scoring of this scale involves adding up the patient’s responses according to specific items and dividing them by the number of items in that scale where a mean item score of ≥3 is considered a level of having distress deserving clinical attention; a score of <2.0 indicates little or no distress, whereas anything in between (score of 2.0 to 2.9) reflects moderate distress [19]. MDDS-17 made changes to a few of the items from the original. The original Diabetes Distress Scale has four items: physician-related distress, regimen-related distress (RD), emotional burden (EB) and T2DM-related interpersonal distress (IPD). MDDS-17 has high internal consistency (Cronbach’s α = 0.940 and test–retest reliability value = 0.33). The modifications made are as follows: the IPD and RD components were merged as one-factor component known as ‘therapeutic support distress’, item 3 was included in the EB subscale, and item 7 was moved from the EB subscale and included under RD based on exploratory factor analysis [20].

Section 6 utilised the revised Malay version of T2DM-related quality of life (Rv-DQOL) adapted from the Diabetes Quality of Life (DQOL) questionnaire. DQOL was intended to be used for evaluating the quality of life specifically related to T2DM and was made up of three major domains, namely, (i) Diabetes Life Satisfaction (QOL Satisfy), 18 items; (ii) Disease Impact Scale (QOL Impact), 27 items; and (iii) Disease-Related Worries Scale (QOL Worry), 14 items and one general question to reflect self-rating of overall general health. All items in the QOL Satisfy domain are scored on a five-point scale, ranging from 1 (very satisfied) to 5 (very dissatisfied), whereas the items in the QOL Impact and QOL Worry are scored on a five-point scale, ranging from 1 (never) to 5 (all the time); the score was presented as the total of the items of each scale divided by the number of items. A higher average score indicates poorer QOL [21]. Recently, a new Malay version of the instrument was developed, keeping the three main domains of ‘satisfaction’, ‘impact’ and ‘worry’ with similar score scale. Each domain’s redundant questions were removed, leaving the newly revised DQOL (Rv-DQOL) with a total of 13 questions. Cronbach’s α values for each domain ranged from 0.75 to 0.93, indicating good internal consistency, and it was validated for use among adult T2DM patients in Malaysia. A higher score indicates a lower quality of life [22]. All three translated questionnaires used had permission from the authors who modified them.

### 2.3. Statistical Analysis

Data were analysed using the SPSS software version 26.0 (SPSS Inc., Chicago, IL, USA). Numerical data were expressed in median and interquartile range (IQR). The dependent variable was sleep quality. Univariate logistic regression analysis was conducted to identify significant factors affecting sleep quality. Any factors with *p* values <0.25 were subjected to multiple logistic regression analyses. A confirmatory analysis to determine the association between sleep quality and quality of life was measured using simple linear regression, in which the dependent variable is quality of life, whereas the independent variable is sleep quality. For all conducted statistical analyses, the significant value was set to α = 0.05.

## 3. Results

### 3.1. Socio-Demographic and Medical Characteristics among Participants

A total of 350 patients with T2DM participated in this study with a response rate of 100%. The median (IQR) age of the respondents was 62.0 (11.0) years. Their age ranged from 28 to 83 years old. The respondents consisted of an equal number of men and women. The majority are Malays, 317 (90.6%) and married (85.4%). One-third of the participants, 32% (95% CI = 27.1, 36.9), had poor sleep quality represented by 112 participants. Table 1 presents the detailed socio-demographic information of the participants.

The median and IQR for HbA1c (%) and BMI in kg/m^2^ were 8.20 (3.20) and 28.05 (6.10), respectively. Table 2 presents medical characteristics and diabetes-related distress among the participants.

### 3.2. Associated Factors for Poor Sleep Quality among Patients with T2DM

Simple logistic regression revealed that T2DM complications, namely, nocturia and RLS, together with the EB component of T2DM-related distress, are significant associated factors contributing to poor sleep quality among the participants. These results are presented in Table 3.

Those who had nocturia had 2.04 times the odds of having poor sleep quality compared with those who did not experience it (OR = 2.04; 95% CI = 1.24, 3.35). Those who suffered from RLS had 2.17 times the odds of having poor sleep quality compared with those without (OR = 2.17; 95% CI = 1.32, 3.56). Participants who were emotionally burdened had 2.37 times chances of having poor sleep quality than those who were not emotionally burdened (OR = 2.37; 95% CI = 1.41, 3.98).

According to the multiple logistic regression analysis shown in Table 3, participants with nocturia, RLS and EB had a significant relationship with poor sleep quality. When controlling for RLS and EB, T2DM patients with nocturia had 1.93 times the odds (Adj. OR = 1.93; 95% CI = 1.16, 3.20; *p* = 0.011) of having poor sleep quality than T2DM patients without nocturia. When nocturia and EB were controlled, participants with RLS had 78% more chances (Adj. OR = 1.78; 95% CI = 1.06, 2.99; *p* = 0.031) of having poor sleep quality. When nocturia and RLS were accounted for, those who were emotionally burdened had two times higher chance of having poor sleep quality than those who were not emotionally burdened (Adj. OR = 2.02; 95% CI = 1.17, 3.48; *p* = 0.011).

### 3.3. Association between Sleep Quality and Quality of Life among Patients with T2DM

Based on simple linear regression model, an increase of 1 in the PSQI-M score would increase the score of T2DM Rv-DQOL by 0.37, but there was no statistically significant relationship between the two (*p* = 0.234; 95% CI = −0.24, 0.98). Furthermore, the r^2^ value was very low (0.004), indicating that sleep quality score (PSQI-M) accounts for only 0.4% of the variation in the quality of life (T2DM RV-DQOL).

## 4. Discussion

### 4.1. Prevalence of Poor Sleep Quality

This study found that the prevalence of poor sleep quality based on the PSQI-M score among T2DM patients attending primary health clinics is 32%. This finding is similar to a study conducted in Japan among similar subjects, which found 30.6% in the participants [23]. Data from a large cohort population study in the United Kingdom indicated that 28% of the T2DM patients studied experienced sleep disturbances [24]. A study conducted by Lou et al. (2015) in China involving 944 patients with T2DM demonstrated that 33.6% of them were poor sleepers [25]. These studies all showed similar results of prevalence: roughly one-third of the T2DM study participants had poor sleep quality. This is supported by a recent meta-analysis, which found that the pooled prevalence of insomnia among T2DM patients was 39% [26]. Nevertheless, the prevalence of poor sleep quality within a control group in a study by Narisawa et al. (2017) was reported to be 38.4% [27]. In a large general population study in China by Tang et al. (2017), the overall prevalence of insomnia was reported to be 26.6% [10]. Poor sleep quality is as common among those without T2DM, signifying the multifactorial attributes of the problem. Inadequate sleep or poor sleep quality has been related to adverse health outcomes such as elevated systemic markers of inflammation, high blood pressure, impaired glucose tolerance and T2DM development [28,29,30,31]. This relationship helps explain the plethora of sleeping problems among T2DM patients.

### 4.2. Associated Factors for Poor Sleep Quality

This study discovered three factors to be significantly associated with poor sleep quality, namely, nocturia, RLS and the EB. A study conducted among female T2DM patients in Taiwan obtained similar finding where nocturia was found to be significantly correlated with sleep quality (Pearson’s correlations showing Chinese PQSI score r = 0.28, *p* < 0.01) [32]. Another study conducted by researchers from a neighbouring country, Singapore, found a significant association between nocturia and poor sleep quality (adjusted prevalence rate ratio of 1.54, 95% CI 1.06–2.26) among the elderly with T2DM, hyperlipidaemia and hypertension [33]. Conversely, a study by Nasseri et al. (2015) found that nocturia did not significantly cause poor sleep quality and concluded that although T2DM is responsible for disrupting sleep, it is unrelated to chronic complications such as nocturia and painful neuropathy [34]. Interestingly, according to a study conducted in Brazil, having nocturia increased the chances of falling into the ‘oversleeping’ (>8 h) category [35]. A logical explanation from our finding is that waking up from sleep to urinate at least once potentially causes cumulative sleep deprivation, more so if patients had to wake up more than once. In addition, waking up in the middle of a deep sleep may negatively affect sleep quality [36].

Numerous studies highlighted that RLS is more common in patients with T2DM compared with the general population, with a prevalence of 8–45% [37,38,39,40,41]. In those without T2DM, the prevalence of having RLS is lower (10–12%) as depicted in studies by Winter et al. (2012) and Zobeiri et al. (2014) [37,42]. In our study, 26% of the participants had symptoms of RLS, and they were significantly associated with poor sleep quality. This is supported by a study from India, where T2DM patients with RLS had a significant delay in sleep onset and a higher PSQI score [43]. Another study in Iran found that T2DM patients who had RLS were nearly three times more likely to have poor sleep quality [39]. RLS is well understood to be a complicated and distressing condition. It disrupts the initiation and subsequent maintenance of sleep because the unpleasant sensation, if present, is more prominent during rest or when lying down, particularly at night. To break free from this, the sufferer must move his/her lower limbs by shaking, walking or stretching. An interventional study from Japan found a significant improvement in RLS symptoms and sleep quality after 3 months of treatment with the dopamine agonist pramipexole [38]. Given the predominance of RLS in T2DM patients and the fact that it is treatable, it is critical to recognise and actively seek it out to improve sleep quality and overall outcomes.

Our study also found that emotionally burdened patients were significantly associated with poor sleep quality. Several local studies used similar tool (MDDS-17) to investigate the prevalence and associated factors of T2DM-related distress; however, to the best of our knowledge, none studied its dependency on sleep quality [44,45,46,47]. A study conducted in Iran demonstrated that psychological distress is an independent risk factor for poor sleep quality among T2DM patients [48]. Two studies from China with similar backgrounds investigated the reverse relationship and revealed that sleeping time is significantly related to both the overall T2DM distress score and its EB component. However, they used total sleep time in a day as a sleep quality measure rather than the PSQI [49,50]. Having disease-related psychological distress and emotional disturbance affect sleep because of interruption in complex neurohormonal physiology could be possible explanations for our findings. Cortisol is released in response to stress, causing agitation and arousal and thus disrupting the sleep–wake cycle, which is controlled by the circadian rhythm.

### 4.3. Association between Sleep Quality and Quality of Life

Using simple linear regression analysis, we discovered a weak and insignificant link between sleep quality and quality of life. Because the r^2^ value was low (0.004), it meant that there were many other factors influencing the variation in the quality-of-life score aside from sleep. The tool we used in this study is relatively new (released in 2018) and has seen limited research use. This tool was used in two local studies that investigated the link between medication adherence and quality of life [51,52]. To the best of our knowledge, this is the first study investigating the relationship between sleep quality and quality of life using the aforementioned tool.

A few local studies investigated the factors that influence the quality of life among T2DM patients using various tools, such as WHO Quality of Life–Brief (WHOQOL-BREF), Short Form Survey 36 (SF-36) and Malay version of DQOL [44,53,54]. A study in the United Arab Emirates found a high independent association between sleep quality and quality of life using WHOQOL-BREF (OR = 8.20, 95% CI 4.34–15.45, *p* < 0.001), which contradicted our findings [17]. A study by Zhang et al. (2016) discovered a significant positive correlation between sleep quality (PSQI) and quality of life (as measured by the Diabetes Specificity Quality of Life Scale (DSQL)) as well as a significant worsening of the DSQL score when poor sleep quality and depression were combined [55]. Poor sleep quality would logically have a detrimental impact on an individual’s ability to function; therefore, they are expected to perceive lower quality of life. Because of the differences in the tools used, our study is unique and not comparable to the other studies stated. Our PSQI cut-off point for defining poor sleep quality is greater than 5, whereas the Chinese version is greater than 7 [56], and Rv-DQOL does not give a cut-off point for distinguishing good and poor quality of life. Furthermore, both sleep and life qualities are subjective; thus, their quantification for the purpose of the analysis is debatable.

Quality of life associates among patients with T2DM is widely studied across the globe. Age and female gender were found to be significant associated factors across different regions based on studies conducted by Rwegerera et al. (2018) in Africa and Rodriguez et al. (2018) in Europe [57,58]. Poor glycaemic control predisposed to the development of complications, and this remained the most common significant determinant of quality of life among patients with T2DM regardless of ethnicity and region [57,58,59].

### 4.4. Socio-Demographic Factors and Sleep Quality

From the results of this study, no socio-demographic factors are significantly related to poor sleep quality among T2DM patients. The mean age of the study participants is 60.6 (0.49) years and is not a significant predictor of poor sleep quality. This outcome is contrary to a few other studies in participants with a mean age of >60 years, which demonstrated that the age factor is significantly related to poor sleep quality, especially for younger ones [23,60]. A study by Shamshirgaran et al. (2017) revealed that middle-aged (50–59-year-old) patients have a significant association with poor sleep quality [48]. Being older predisposes one to changes in sleep architecture and worsening of physical conditions, all of which contribute to a reduction in sleep quality. However, in our settings, particularly in this state, Kelantan, one of Malaysia’s most laid-back states, is a preferable settling-down place for locals, particularly those aged >60 years. They are content in their motherland and therefore have gained inner peace.

In our study, the percentage of having poor sleep quality is comparable in both sexes (women 57% vs. men 55%); however, it did not play a role in determining sleep quality. This finding is supported by a study in Japan involving a control group [27]. Contrarily, other studies in the literature outlined being female as a significant association factor [61,62,63], and another study pointed out that male sex was associated with poor sleep quality [39].

Employment status did not have a role in causing poor sleep quality in our study, which was supported by a study from Japan conducted by Narisawa et al. (2017) [27]. In contrast, a study found that unemployed people were significantly more likely to have poor sleep quality than those who worked full time [63]. In this study, the educational level among participants did not influence their sleep quality. Most of our participants (71.7%) did not pursue tertiary education. This is in line with a study conducted in Iran by Barakat et al. (2019) [63]. Another study by Zhang et al. (2016) also demonstrated that educational level does not influence the quality of life of T2DM patients [55].

A study by Modarresnia et al. (2018) reported that being single is a significant factor for poor sleep quality (OR 3.6, *p* < 0.005) [39], whereas our study showed that marital status has no predilection for sleep quality, which is supported by few other studies [17,63].

### 4.5. Medical Characteristics and Sleep Quality

In our study, diagnosis duration of more than 10 years, treatment with insulin, presence of painful neuropathy, HbA1c level and BMI did not contribute to poor sleep quality. In contrast, a study by Nasseri et al. (2015) found that having T2DM for more than 10 years was associated with poor sleep quality, with an OR of 2.63 and a *p* value of 0.02 [34]. Another study found that people who had T2DM for more than 6 years had 1.8 times the likelihood of having poor sleep quality, with a *p* value of 0.003 [48]. A study by Bani-Issa et al. (2018) agreed with us on this matter, showing that the duration of having T2DM of >10 years was not significantly related to poor sleep quality [17]. They differed from us by having a minority number of patients with DM duration of >10 years (25%), whereas our study had almost double the portion of those (49.1%). The longer duration of the disease is logically related to more complications that may likely affect sleep quality. Our participants have almost equal proportions among those having diabetes for more than 10 years and less (49.1% vs. 50.9%). This might not be accurate, as most T2DM patients seek medical attention much later after the onset of the disease, especially when they start to have symptoms. This variability could explain the lack of association between duration of T2DM diagnosis and sleep quality.

Our study did not show that insulin use is related to poor sleep quality. This is contrary to a study demonstrating that insulin use significantly increases the chance of having poor sleep quality by 2.17 times [63]. A study by Modarresnia et al. (2018) supported our finding that there is no significant difference in sleep quality among those treated with insulin [39].

Our study highlighted that uncontrolled DM (HbA1c ≥ 7%) did not significantly affect sleep quality. In our study, the median HbA1c was 8.2% and the IQR was 3.2%. The minority (26.3%) of the participants had HbA1c of 7% and more, which might explain the lack of association in our study. A study conducted by Nasseri et al. (2015) with mean HbA1c among participants (7.1 ± 1.47)%, and 53% with good glycaemic control supported our finding [34]. Another study by Tsai et al. (2012) showed a positive correlation between sleep quality and HbA1c (r = 0.54, *p* < 0.01) and explained postulated pathophysiology relating to cortisol release (which increased blood sugar level) in response to poor sleep quality [64]. A meta-analysis revealed that short sleep duration was significantly associated with increased HbA1c, and the finding that good sleep quality improved fasting blood glucose supports this idea [65].

In our study, we classified patients with BMI ≥ 23 kg/m^2^ as overweight according to the WHO expert consultation on the appropriate body mass index for Asian populations. Asian populations have different health risk profiles than European populations; thus, BMI of ≥23 kg/m^2^ was classified as an increased risk [66]. In our study, 91.7% of the patients were overweight, but they had no predilection of having poor sleep quality. A study by Shamshirgaran et al. (2017) found that having a BMI of more than 25 kg/m^2^ was not linked to poor sleep quality [48]. Another study by Yoshikawa et al. (2022) found a significant correlation between BMI and PSQI scores [60]. A study by Narisawa et al. (2017) discovered an association between a greater BMI and lower sleep quality [27]. Both studies with opposite findings to us attributed the possibility of OSA, which is common among patients with T2DM, causing sleep fragmentation due to intermittent hypoxia, which would subsequently affect sleep quality.

Diabetic neuropathy is a common microvascular complication among T2DM patients and has a wide range of symptoms. Some are painful, whereas others are asymptomatic yet present with a full-blown neurological deficit on examination [67]. About 15–20% of T2DM patients experience neuropathic pain, which is linked to mood and sleep disorders. There is a bidirectional link between neuropathic pain and sleep problems [68]. Patients with neuropathic pain are more likely to develop sleep disturbances, and the lack of sleep and/or poor sleep quality exacerbates the pain [69]. From our subjective, self-administered questionnaire on painful neuropathy, 44.9% of the participants claimed to have it; however, it has no statistical significance in sleep quality. This condition may be overdiagnosed without proper clinical examination and objective assessment by a nerve conduction study, posing a limitation to our study. A study from China with a similar subjective assessment supported our findings [70]. A large national study from Korea with a more objective assessment of diabetic neuropathy using the Michigan Neuropathy Screening Instrument and 10 g monofilament (5.07 Semmes–Weinstein) test found a significant association with sleep impairment [67].

### 4.6. Limitations of Our Study

Our study has several limitations. First, this is a cross-sectional study; hence, we are unable to evaluate the actual causes of poor sleep quality and its relation to the quality of life. Second, our study was conducted in hospital-based primary care, and due to geographical factors, the majority were of Malay ethnicity (90.6%). Both of these factors do not represent the general population of Malaysia, which is a multiracial nation, where Chinese and Indians also contribute to a significant proportion of the population. In addition, primary care services in the hospital are limited in Malaysia and are usually available in university and private settings. Third, there was no control group in this study for a more valid comparison between the factors being studied. Fourth, sleep quality was subjectively assessed using a self-administered questionnaire without objective confirmation with a polysomnograph that could pick up OSA or actigraph that could record actual sleep duration. Fifth, there is a lack of information on therapeutic schemes among participants. As a result, we might have overlook other potential causes of nocturia apart from hyperglycaemia, e.g., sodium-glucose cotransporter-2 inhibitor (SGLT-2 i). We recommend further study to look at other causes of nocturia (e.g., prostatism) and its relationship with poor sleep quality. In addition, this study did not identify participants by the number of insulin injections and looked at their relationship with sleep quality. Hence, we recommend further study to determine the relationship between numbers of insulin injections or dosages or type of regime with sleep quality and quality of life. Lastly, we did not include important social backgrounds or established psychiatric problems that may affect sleep quality and quality of life.

## 5. Conclusions

In summary, this study found that 32% of our studied population have poor sleep quality. Nocturia, restless leg syndrome and the EB component of type II diabetes (T2DM) distress are factors significantly associated with poor sleep quality. However, there is a lack of association between sleep quality and quality of life. Patients with T2DM should be managed holistically, especially at the primary care level. Treating doctors should actively evaluate the severity of diabetic complications as objectively as possible based on available resources. Another important aspect is improving sleep. Thus, comprehension and sleep hygiene education are critical to achieving better overall outcomes.

## Figures and Tables

**Table 1 ijerph-19-05211-t001:** Socio-demographic characteristics of participants (*n* = 350).

Variables	Total	Poor Sleep	Good Sleep
	*n* (%)	*n* (%)	*n* (%)
Age (years) ^a^	62.00 (11.00)	61.00 (12.00)	62.00 (10.00)
Sex (Female/Male)	175/175 (50.0/50.0)	57/55 (50.9/49.1)	118/120 (49.6/50.4)
Ethnicity (Malays/Non-Malays)	317/33 (90.6/9.4)	100/12 (89.3/10.7)	217/21 (91.2/8.8)
Marital Status (Married/Others)	299/51 (85.4/14.6)	98/14 (87.5/12.5)	201/37 (84.5/15.5)
Education (<Tertiary/Tertiary Onwards)	251/99 (71.7/28.3)	81/31 (72.3/27.7)	170/68 (71.4/28.6)
Working Status (Working/Not Working)	121/229 (34.6/65.4)	41/71 (36.6/63.4)	80/158 (33.6/66.4)

^a^ Median (IQR).

**Table 2 ijerph-19-05211-t002:** Medical characteristics and diabetes-related distress among participants (*n* = 350).

Variables	Total	Poor Sleep	Good Sleep
	*n* (%)	*n* (%)	*n* (%)
**Medical Characteristics**			
Duration of Diagnosis as DM (<10/≥10 (years))	178/172 (50.9/49.1)	59/53 (52.7/47.3)	119/119 (50/50.0)
Insulin Therapy (Yes/No)	160/190 (45.7/54.3)	55/57 (49.1/50.9)	105/133 (44.1/55.9)
Painful Neuropathy (Yes/No)	157/193 (44.9/55.1)	56/56 (50/50.0)	101/137 (42.4/57.6)
Nocturia (Yes/No)	222/128 (63.4/36.6)	83/29 (74.1/25.9)	139/99 (58.4/41.6)
Restless Leg Syndrome (Yes/No)	91/259 (26/74.0)	41/71 (36.6/63.4)	50/188 (21/79)
HbA1c (%) ^a^	8.20 (3.20)	8.00 (3.30)	8.20 (3.20)
HbA1c (Controlled (<7%)/Uncontrolled (≥7%))	258/92 (73.7/26.3)	82/30 (73.2/26.8)	176/62 (73.9/26.1)
BMI (kg/m^2^) ^a^	28.05 (6.10)	28.48 (5.50)	28.00 (5.90)
BMI (≥23/<23 (kg/m^2^))	321/29 (91.7/8.3)	106/6 (94.6/5.4)	215/23 (90.3/9.7)
**Diabetes-Related Distress**			
Emotional Burden (No/Yes)	272/78 (77.7/22.3)	75/37 (67/33.0)	197/41 (82.8/17.2)
Physician related Distress (No/Yes)	334/16 (95.4/4.6)	106/6 (94.6/5.4)	228/10 (95.8/4.2)
Therapeutic Support Distress (No/Yes)	319/31 (91.1/8.9)	98/14 (87.5/12.5)	221/17 (92.9/7.1)

^a^ Median (IQR).

**Table 3 ijerph-19-05211-t003:** Associated factors for poor sleep quality among patients with T2DM using simple and multiple logistic regression.

Variables	Crude OR ^a^ (95% CI) ^b^	*p* Value	Adj. OR ^c^ (95% CI) ^b^	*p* Value
**Sociodemographic**				
Age (years)	1.00 (0.98–1.03)	0.978		
Sex (Female/Male)	1.00/1.05 (0.67–1.65)	0.819		
Ethnicity (Malays/Non-Malays)	1.00/0.81 (0.38–1.70)	0.573		
Marital Status (Married/Others)	1.00/1.29 (0.67–2.50)	0.452		
Education (<Tertiary/Tertiary Onwards)	1.00/1.05 (0.63–1.72)	0.863		
Working Status (Working/Not Working)	1.00/1.14 (0.71–1.82)	0.583		
**Medical Characteristics**				
Duration of Diagnosis as DM (<10 years/≥10 years)	1.00/0.90 (0.57–1.41)	0.64		
Insulin Therapy (Yes/No)	1.00/1.22 (0.78–1.92)	0.382		
Painful Neuropathy (Yes/No)	1.00/1.36 (0.86–2.13)	0.185		
Nocturia (Yes/No)	1.00/2.04 (1.24–3.35)	0.005 *	1.00/1.93 (1.16–3.20)	0.011 *
Restless Legs Syndrome (Yes/No)	1.00/2.17 (1.32–3.56)	0.002 *	1.00/1.78 (1.06–2.99)	0.031 *
HbA1c (Controlled (<7%)/Uncontrolled (≥7%))	1.00/0.96 (0.58–1.60)	0.884		
BMI (<23 kg/m^2^/≥23 kg/m^2^)	1.00/1.89 (0.75–4.78)	0.179		
**Diabetes-Related Distress**				
Emotional Burden (No/Yes)	1.00/2.37 (1.41–3.98)	0.001 *	1.00/2.02 (1.17–3.48)	0.011 *
Physician related Distress (No/Yes)	1.00/1.29 (0.46–3.64)	0.630		
Therapeutic Support Distress (No/Yes)	1.00/1.86 (0.88–3.92)	0.104		
Overall Diabetes Distress (No/Yes)	1.00/0.94 (0.42–2.07)	0.936		

^a^ Crude Odds Ratio ^b^ Confidence Interval ^c^ Adjusted Odd Ratio * *p* value ≤ 0.05. Note: No significant interaction; no multicollinearity problem; model assumptions met.

## Data Availability

The data presented in this study are available on request from the corresponding author. The data are not publicly available due to privacy and confidentiality.

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
