# Peer review of "Sleep Quality among Patients with Type 2 Diabetes: A Cross-Sectional Study in the East Coast Region of Peninsular Malaysia"

_ijerph, 2022, doi:10.3390/ijerph19095211_

Round 1
Reviewer 1 Report
Dear Authors,
The manuscript entitled “Sleep Quality among Patients with Type 2 Diabetes Mellitus: A Cross-Sectional Study in the East Coast Region of Peninsular Malaysia” deals with an interesting and current topic, and the manuscript is of high-quality. I have only a few minor recommendations to deal with.
Concerning the abstract, it is too detailed; lines 22-24 should be removed, since the last sentence repeats the results. Besides, please, do not use abbreviations in the abstract (especially if you do not explain them, see IQR).
In line 37 you mention 1.6 million deaths worldwide but you do not add the time frame (maybe one year?). The use of footnotes is different in Table 1 and 2 (variables vs. values). In lines 209-210 you mention 1.37 odds ratio, while the corresponding value in Table 4 is 2.37. Table 5 is unnecessary, since all data is mentioned in the text. When discussing the prevalence of poor sleep quality (section 4.1), it would be necessary to compare the prevalence of this condition in T2DM patients to the prevalence in the population not suffering from T2DM; it is theoretically possible that there is no difference among the two subgroups of population. Similarly, concerning RLS, you only mention that it is more common in patients with T2DM compared with the general population, but you provide data only for patients with T2DM (lines 268-269). Please, revise the sentence in lines 329-330. The term “acceptance” in line 367 is incorrect.
There are some typos, e.g., in lines 22 (see the format of confidence intervals), 121, 405, 492, and 585 (unnecessary spaces), 182 (superscript), 248 and 269 (citation format), 288 (repetition of the term “T2DM”; the term “Another” is incorrect).
Reviewer 2 Report
In this manuscript Dr Nor Fareshah Mohd Nasir and coauthors explore a possible correlation between sleep quality in patients with type 2 diabetes mellitus (T2DM) and their quality of life. This study is a cross-sectional survey including 350 participants (175 men and 175 women). To explore quality of life, they use a modified form of PSQI (Malay version of P. Sleep Quality Index, PSQI-M). In a regression analisys authors observe that poor sleep quality is significantly associated with nocturia (odds ratio [OR] 2.04; 95% CI 1.24, 3.35), restless legs syndrome (OR 2.17, 95% CI 1.32–3.56) and emotional burden (OR 2.37, 95% CI 1.41–3.98). Surprisingly, poor quality of sleep does not correlate with quality of life of life of diabetic patients.
This work is interesting because it explores possible correlates of quality of life in patients with diabetes. The leterature provides a lot of data regarding this issue, however further information in different ethinc groups are welcome.
I have some concerns:
1) First of all, delete "mellitus" throughout the text and tables because it is outdated. Worldwide correct definition is type 2 diabetes.
2) Please recreate all tables replacing dycotomous variables. Each row has to include both the conditions, e.g.: sex (male/female) n°, % 175/175 (50/50), and so on
3) An important limitation of this stusy is the lack of information regarding the therapeutic scheme of participants. As a result, authors cannot exclude that nocturia depends on particular drugs such as diuretics or SGTL-2 inhibitors, making the relation between nocturia and poor sleep quality at least questionable. Moreover, to determine whether insulin may influence or not the quality of life and/or quality of sleep authors must show a correlation with the number of insulin Unit taken by partecipants, since it is obvious that patients in basal bolus have an higher risk to be sad and depressed than participants using only basal insulin (4 administration vs only 1).
4) WHO stated that overweight begins with BMI ≥ 25 kg/m2, please edit this disgrace
5) Since poor quality of sleep and and peripheral clock gene expression unbalanced may influence the risk of diabetes and its metabolic control, please comment the following papers in discussion: 1) PMID: 33788000, 2) PMID: 30877388
Round 2
Reviewer 2 Report
Authors Adde esse all my concerns. Thank you